# TG-RAG: A Retrieval-Augmented Framework for Reasoning Guidance in Specialized Domains

Liang Su [*1]  Mingyang Zhang [*2]  Yun Xiong [1]  Tengfei Liu [2]  Siwei Zhang [1]  Xi Chen [1]  Li Sun [3]

## Abstract

Enhancing Large Reasoning Models (LRMs) for specialized domains remains a critical challenge. While recent industrial frameworks attempt to encapsulate Standard Operating Procedures into modular "skills" for dynamic retrieval, utilizing them via context engineering often proves insufficient for complex workflows, leading to "Cognitive Drift." To mitigate this, we propose **Thought Guidance-Retrieval Augmented Generation (TG-RAG)**, a Retrieval-Augmented framework that effectively steers the generation process without relying solely on the model's self-correction. Built upon an Expert Procedure Graph (EPG) that formalizes unstructured SOPs, the framework uniquely employs a dynamic **"Interrupt-Retrieve-Generate" (IRG)** mechanism to actively inject step-specific directives into the model's reasoning process. Extensive evaluations show that TG-RAG achieves competitive performance, demonstrating advantages in specialized domains by ensuring faithful adherence to domain SOPs. Code is available at https://github.com/V1ncent-S/Thought-Guidance

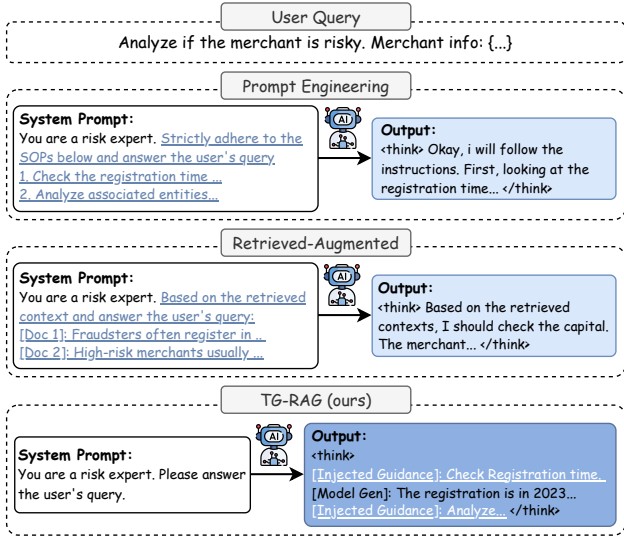

*Figure 1.* Comparison of different guidance mechanisms. While Prompt Engineering and Retrieved-Augmented methods provide static guidance within the query, TG-RAG dynamically injects the guidance into the model's reasoning process.

## 1. Introduction

Large Reasoning Models (LRMs) (Xu et al., 2025), such as Deepseek-R1 (DeepSeek-AI, 2025) and Qwen3 (Qwen Team, 2025), have demonstrated remarkable capabilities by internalizing "Chain-of-Thought" (CoT) (Wei et al., 2022) methodologies. To leverage these capabilities for special-

ized tasks, a prevalent trend in autonomous agents is to encapsulate domain-specific knowledge—such as Standard Operating Procedures (SOPs)—into modular "skills" (e.g., instruction files or scripts) that agents can retrieve and apply (Zhang et al., 2025a; 2026a). However, while this "skill-based" approach effectively packages expertise, executing these skills in high-stakes domains like finance and medicine remains a critical challenge. When SOPs involve intricate branching logic or extended reasoning chains, models frequently succumb to "*Cognitive Drift*" (Zhang et al., 2026b)—a deviation from the intended expert protocol. In such safety-critical contexts, even minor logical errors can lead to severe consequences, making strict procedural fidelity non-negotiable (Peng et al., 2024).

Efforts to mitigate cognitive drift have traditionally relied on prompt engineering (Sahoo et al., 2024; Kojima et al., 2022; Wang et al., 2023b) and model fine-tuning (Rafailov et al., 2023; Shao et al., 2024; Yu et al., 2025). While sophisticated prompting can provide an initial static plan (Wang et al., 2023b; Yue et al., 2025), the guiding influence of a

*Equal contribution [1] College of Computer Science and Artificial Intelligence, Fudan University, Shanghai, China [2] Ant Group, Hangzhou, China [3] School of Computer Science, Beijing University of Posts and Telecommunications, Beijing, China. Correspondence to: Liang Su <lsu23@m.fudan.edu.cn>, Yun Xiong <yunx@fudan.edu.cn>.

*Proceedings of the 43rd International Conference on Machine Learning*, Seoul, South Korea. PMLR 306, 2026. Copyright 2026 by the author(s).

static context often diminishes over long, complex reasoning trajectories. Conversely, fine-tuning on expert trajectories is resource-intensive and rigid; it struggles to adapt to the rapidly evolving nature of domain knowledge and risks compromising the model's general reasoning capabilities.

Consequently, Retrieval-Augmented Generation (RAG) (Lewis et al., 2020; Gao et al., 2023) has emerged as a flexible, training-free alternative. By interfacing with external sources at inference time, RAG theoretically offers a mechanism for real-time intervention. The evolution of this paradigm reflects a clear progression: from retrieving declarative facts to fetching structured guidance, such as CoT exemplars (Li et al., 2025) or graph-based reasoning paths (Wu et al., 2025b; Zhao et al., 2024; Wang et al., 2025). This trajectory has culminated in the Retrieval-Augmented Thought (RAT) (Wang et al., 2024b; Zhang et al., 2025b) approach, which aims to actively influence the reasoning process itself (Wang et al., 2023a; Trivedi et al., 2023).

Despite this progress, a fundamental integration gap persists. Existing methods typically deliver retrieved SOPs or skills as passive contextual information within the prompt. This indirect form of "suggestion" lacks enforceability, leaving the model susceptible to ignoring instructions or hallucinating steps during multi-step reasoning.

To bridge this gap, we introduce **T**hought **G**uidance-**R**etrieval **A**ugmented **G**eneration (**TG-RAG**), a reasoning-time retrieval framework designed to robustly steer LRMs through expert-defined workflows. Rather than relying on unstructured text skills, we propose formalizing these SOPs into an *Expert Procedure Graph (EPG)*—a structured knowledge base that models the conditional logic of expert protocols. At each reasoning juncture, TG-RAG employs a dynamic "*Interrupt-Retrieve-Generate (IRG)*" mechanism. This mechanism identifies the model's current state within the EPG and injects the specific directive for the next step directly into the model's active reasoning stream.

We term this steering mechanism *Thought Guidance (TG)*. Unlike conventional prompting that offers a static plan, or standard RAG that provides passive context, TG operates as a proactive, in-process intervention, as illustrated in Figure 1. By breaking complex SOPs into atomic, executable directives and injecting them sequentially, TG-RAG significantly reduces the cognitive load on the model and mitigates the risk of drift.

Comprehensive experimental results demonstrate that TG-RAG achieves significant performance improvements over existing methods, particularly in specialized domains.

Our TG-RAG framework is entirely *training-free* and highly *compatible* across diverse LRMs. The contributions of this work are as follow:

- We introduce TG-RAG, a reasoning-time framework that employs the IRG mechanism to actively steer model generation. This shifts SOP integration from passive context instruction to dynamic, in-process guidance.
- We design the Expert Procedure Graph (EPG), a hierarchical knowledge base tailored for modeling procedural knowledge from specialized domain SOPs.
- We validate TG-RAG across specialized domains, demonstrating that it consistently outperforms strong baselines in both reasoning accuracy and procedure adherence.

## 2. Related Work

### 2.1. Retrieval-Augmented Generation

Standard RAG (Lewis et al., 2020; Gao et al., 2023) enhances LLMs by retrieving factual information to enrich the initial context. Recent advancements have evolved this paradigm by enabling iterative retrieval (Asai et al., 2024; Shao et al., 2023) or moving beyond facts to fetch structured guidance like plans (Wu et al., 2025b; Wang et al., 2025). However, these approaches typically provide guidance as a static, upfront context.

A prominent recent advancement in this second direction is the BoT framework (Yang et al., 2024), which retrieves thought-templates to structure the model's reasoning. However, frameworks like BoT operate primarily at the prompt level. The retrieved template is provided as a static upfront context, relying on the model's instruction-following capabilities to execute the blueprint holistically.

A more dynamic paradigm, Retrieval-Augmented Thought (RAT) (Wang et al., 2024b), integrates retrieval directly into the reasoning loop. This includes modifying or correcting a previously generated thought (e.g., KD-CoT (Wang et al., 2023a)), selecting the most promising reasoning path from multiple candidates (Wang et al., 2024b; Zhang et al., 2025b), or retrieving fresh knowledge based on reasoning output to better inform the very next step (Trivedi et al., 2023).

Despite these advancements, a critical gap persists: existing methods deliver guidance as passive context. TG-RAG addresses this by proactively interrupting generation to inject explicit, step-specific directives, shifting the paradigm from passive contextual suggestion to active, enforceable procedural steering.

### 2.2. Controllable Reasoning Steering

Guiding LLM reasoning is a central challenge in controllable generation. A primary approach is static, upfront control via prompt engineering (Sahoo et al., 2024; Yao et al., 2022; Kojima et al., 2022; Wang et al., 2023c). Methodologies like CoT (Wei et al., 2022) and Plan-and-Solve (Wang

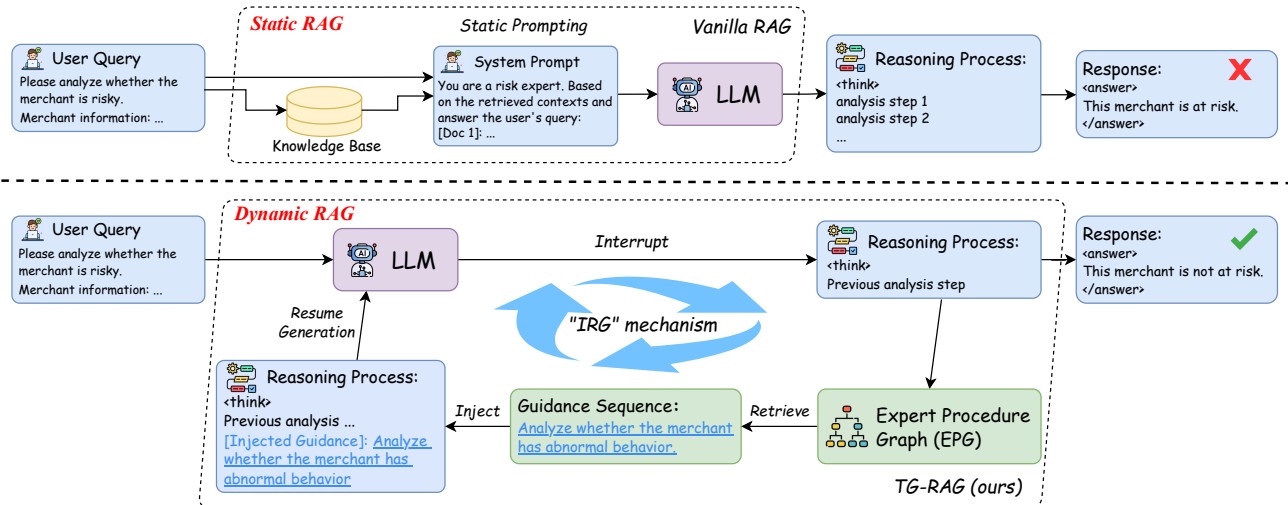

*Figure 2.* Overview of the TG-RAG Framework. The framework employs an "IRG" mechanism to dynamically retrieve procedural guidance from the EPG. This guidance is injected directly into the model's reasoning process (highlighted in blue) to steer the process, contrasting with static RAG approaches.

et al., 2023b) structure the problem-solving process externally but lack adaptability once generation begins, often struggling to maintain logical coherence over long horizons. More dynamic control is offered by search-based methods like ToT (Yao et al., 2023) and GoT (Besta et al., 2024), which explore multiple paths but incur significant computational overhead. A more fine-grained category involves intervening in the generation process itself (Zou et al., 2023), including "Thought Intervention" (Wu et al., 2025a) and activation steering methods (Wang et al., 2024a; Højer et al., 2025; Zhang et al., 2024), which largely operate by reactively correcting errors or manipulating latent states rather than enforcing explicit procedural logic.

TG-RAG distinguishes itself by being proactive and structurally grounded. Instead of correcting errors after they occur or relying on implicit steering, it utilizes the EPG to impose a topological constraint that guides the reasoning trajectory from the outset.

## 3. Methodology

The TG-RAG framework is designed to enforce the rigorous execution of Standard Operating Procedures—increasingly packaged as Agent Skills—within LRMs. As illustrated in Figure 2, the framework operates as a dynamic closed-loop system that augments standard generation with real-time procedural constraints. Specifically, TG-RAG integrates two core components: (1) the Expert Procedure Graph (EPG), a structured knowledge base that formalizes the logic inherent in expert skills; and (2) the "Interrupt-Retrieve-Generate" (IRG) mechanism, which actively aligns the model's reasoning trajectory with the EPG by dynamically injecting step-wise directives.

### 3.1. Preliminary

We formulate the task of controllable reasoning as a conditional generation problem. Let $\mathcal{M}$ be an LRM parameterized by $\theta$. Given an input $\mathbf{x}$, the model generates an output sequence $\mathbf{y}$, which is typically partitioned into an intermediate reasoning chain $\mathbf{r}$ and a final answer $\mathbf{a}$, denoted as $\mathbf{y} = [\mathbf{r}, \mathbf{a}]$. The generation follows an auto-regressive probability distribution: $P_\theta(\mathbf{y}|\mathbf{x}) = \prod_t P_\theta(y_t|\mathbf{x}, y_{<t})$.

Conventional prompting and RAG methods attempt to guide this process by augmenting the input space. They construct a refined context $\mathbf{x}' = [\mathcal{I}, \mathcal{K}, \mathbf{x}]$, where $\mathcal{I}$ and $\mathcal{K}$ represent static instructions and retrieved knowledge, respectively. However, the subsequent reasoning $\mathbf{r}$ remains governed solely by the model's internal priors. In contrast, TG-RAG intervenes directly within the probability space of the reasoning process $\mathbf{r}$. At discrete logic junctures, the framework retrieves a targeted guidance sequence $\mathbf{g}$, transforming the generation condition to $P_\theta(\mathbf{r}_t|\mathbf{x}, \mathbf{r}_{<t}, \mathbf{g})$. This ensures the generation trajectory is strictly conditioned on the expert-defined procedure in real-time.

### 3.2. The Expert Procedure Graph

To effectively steer LRMs through specialized tasks, we propose a fundamental design principle: the decoupling of procedural logic from domain knowledge. Unlike existing agent frameworks that encapsulate skills as unstructured XML file folders—where logic and facts are intertwined—we formalize the "Expert Skill" into a structured Expert Procedure Graph (EPG). For a given class of specialized tasks, the underlying compliance rules and workflows are universal and highly stable. Thus, the EPG is established

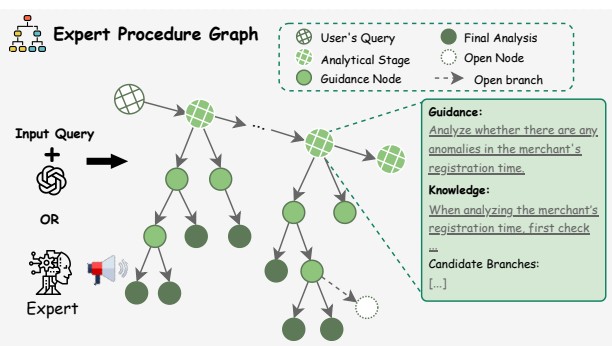

*Figure 3.* Illustration of the Chain-of-Trees structure of EPG.

as a persistent topological blueprint that accurately reflects the invariant nature of these expert protocols, avoiding the structural unpredictability of dynamically generated paths.

Formally defined as a directed acyclic graph $G = (V, E)$, the EPG explicitly models the Standard Operating Procedure as the graph's topology (Edges $E$), while encapsulating domain-specific contexts within the nodes (Vertices $V$). Specifically, we define each node $v$ as a tuple $(\mathbf{g}_v, \mathcal{C}_v)$. Here, $\mathbf{g}_v$ represents the "*Action Directive Guidance*", serving as the invariant step instruction. The component $\mathcal{C}_v$ functions as a flexible Knowledge Container. Crucially, this container is designed as an agnostic interface: it can either house fundamental domain knowledge, or seamlessly integrate with a RAG system to retrieve real-time evidence. This design ensures that the node provides not just the "instruction" of what to do, but also the precise "information" required to do it.

Architecturally, the EPG draws inspiration from decision tree algorithms to explicitly model the conditional nature of expert problem-solving. It adopts a "*Chain-of-Trees*" topology: the macro-level workflow is represented as a sequential chain of Analytical Stages, while the micro-level reasoning within each stage is encoded as decision sub-trees, as shown in Figure 3. This decoupled forest structure provides inherent error isolation: an incorrect routing decision in one sub-tree is confined to that specific local analysis, preventing the error from cascading into independent logic branches. Additionally, we explicitly embed terminal reflection nodes at the end of key EPG trajectories, forcing the model to conduct a final self-correction pass against the initial query to rectify preceding routing errors. Reflecting the rigorous standardization of professional domains, the EPG is designed as a rooted graph with a fixed entry point. This topology ensures that the model strictly adheres to the necessary initialization protocols, preventing the omission of foundational diagnostic steps that often occurs in free-form generation.

This decoupling strategy significantly streamlines the con-

struction and maintenance of the knowledge base. Since the graph structure solely represents the logical skeleton of the SOP—which is generally invariant—we employ a semi-automated construction pipeline. We emphasize that specialized SOPs (e.g., medical guidelines) already feature explicit conditional logic, unlike generic unstructured documents. Thus, the LRM primarily performs a deterministic topological mapping rather than open-ended knowledge extraction, inherently minimizing structural hallucinations.

An LRM first acts as a parser to extract the procedural structure from raw domain documents, generating a candidate graph. Domain experts are then tasked only with structural verification (i.e., checking the correctness of decision branches) rather than laborious content authoring. Empirical evidence from our experiments indicates that this approach reduces expert involvement to approximately 30 minutes per domain.

Importantly, this is a one-time upfront cost. Because SOPs address static standardized workflows, this 30-minute verification effort is effectively amortized across massive-scale subsequent inferences with zero additional human intervention. (See Appendix B for the detailed construction procedure and a complete real-world EPG example).

Furthermore, the *Knowledge Container* mechanism ensures long-term maintainability through hot-swappable updates. For fast-evolving domains, one can simply update the documents in the external RAG knowledge base. Since the container leverages real-time retrieval, the system instantly reflects the latest domain facts without requiring reconstruction of the EPG topology.

### 3.3. The Interrupt-Retrieve-Generate Mechanism

The Interrupt-Retrieve-Generate (IRG) mechanism serves as the dynamic inference engine that operationalizes the static EPG structure. Unlike standard RAG frameworks that passively retrieve context at the input stage, IRG establishes a state-dependent control loop that actively orchestrates the interplay between the model's intrinsic generation and external procedural constraints. This mechanism functions by interrupting the continuous generation stream at discrete reasoning junctures, determining the topologically valid next-step guidance, and injecting it to steer the subsequent trajectory. The complete execution flow is formalized in Algorithm 1.

#### 3.3.1. IRG TRIGGERING PIPELINE.

To operationalize this active intervention, the IRG framework must precisely determine when and how to interrupt the continuous generation process. This is achieved through a three-step pipeline. First, the LRM is explicitly instructed via system prompts to encapsulate each discrete logical de-

**Algorithm 1** The IRG Mechanism

---

**Require:** User Query $\mathbf{x}$, EPG $\mathcal{G} = (\mathcal{V}, \mathcal{E})$
**Ensure:** Final Reasoning Chain $\mathbf{r}$

1: **Initialize:** $\mathbf{r}_0 \leftarrow \emptyset$, $v_0 \leftarrow \text{Root}(\mathcal{G})$, step $t \leftarrow 0$
2: **while** $v_t$ is not a terminal node **do**
3:     *# Phase 1: State-Dependent Semantic Routing*
4:     Let $\mathcal{N}(v_t)$ be the set of valid outgoing neighbors of $v_t$
5:     **if** $|\mathcal{N}(v_t)| > 1$ **then**
6:         *# Select optimal branch or fallback to Open Branch*
7:         $v_{t+1} \leftarrow \text{SemanticRouter}(\mathbf{r}_t, \mathcal{N}(v_t))$
8:     **else**
9:         $v_{t+1} \leftarrow \text{SingleChild}(v_t)$
10:     **end if**
11:     *# Phase 2: Dual-Pathway Injection*
12:     Extract tuple $(\mathbf{g}_{v_{t+1}}, \mathcal{C}_{v_{t+1}})$ from node $v_{t+1}$
13:     *# Condition logic on Action Directive Guidance*
14:     $\mathbf{r}'_t \leftarrow [\mathbf{r}_t, \mathbf{g}_{v_{t+1}}]$
15:     *# Augment input with Knowledge*
16:     $\mathbf{x}' \leftarrow [\mathbf{x}, \mathcal{C}_{v_{t+1}}]$
17:     *# Phase 3: Generation & Physical Interruption*
18:     $\mathbf{s}_{t+1} \leftarrow P_\theta(\cdot \mid \mathbf{x}', \mathbf{r}'_t)$
19:     *# Generate step sequence until a procedural boundary is reached*
20:     **Interrupt** the generation when `</analysis>` is emitted.
21:     **Update:** $\mathbf{r}_{t+1} \leftarrow [\mathbf{r}'_t, \mathbf{s}_{t+1}]$
22:     $t \leftarrow t + 1$
23: **end while**
24: **return** $\mathbf{r}$

---

duction within specific structural tags (i.e., `<analysis>` and `</analysis>`, see Figure 6). Second, we leverage the sequence interruption capabilities of the underlying inference engine (e.g., vLLM) to actively pause generation the exact moment the closing `</analysis>` tag is emitted. Finally, during this physically suspended state, the system invokes the semantic router to evaluate the generated thought against the EPG, retrieves the appropriate procedural directive, injects it into the context, and resumes the generation.

### 3.3.2. STATE-DEPENDENT SEMANTIC ROUTING

The core challenge in traversing the EPG lies in determining the optimal path when encountering nodes with mutually exclusive branches. We formulate this process not as a traditional vector similarity search, but as a semantic routing problem. Building upon the suspended state mentioned above, the framework tasks the LRM to evaluate its own just-generated analysis against the candidate branches defined by the current EPG node (see Figure 7). By leveraging the

LRM's semantic understanding, we transform procedural pathfinding into a context-aware classification task, ensuring that the transition to the next node is logically consistent with the ongoing reasoning context.

To address the theoretical concern that a finite EPG cannot exhaustively enumerate all real-world scenarios—the problem of SOP Incompleteness—we incorporate an Open Branch mechanism as a robust fallback strategy. This "shadow edge" allows the system to handle "*Out-of-Distribution*" cases where pre-defined expert paths are inapplicable, reverting temporarily to the model's intrinsic reasoning capabilities (see Figure 8). Empirical analysis in our experiments indicates that this fallback is triggered in less than 1% of cases, suggesting that while the Open Branch provides a critical safety net for resilience, the EPG structure successfully covers the vast majority of standard workflows.

### 3.3.3. DUAL-PATHWAY INJECTION

Once the appropriate target node is identified, integrating its encapsulated information requires a sophisticated strategy to prevent signal dilution. Building upon our foundational decoupling principle, we employ a Dual-Pathway Injection strategy that processes the node's distinct components through separate, specialized channels. Formally, let $\mathbf{r}_{<t}$ denote the current reasoning trajectory and $\mathbf{x}$ represent the initial user query.

The content derived from the Knowledge Container ($\mathcal{C}_v$)—whether it comprises static domain annotations or dynamic, real-time evidence retrieved via external RAG—is seamlessly injected into the input context. This forms an enriched input representation $\mathbf{x}' = [\mathbf{x} \oplus \mathcal{C}_v]$, which acts purely as an informational signal to provide the necessary factual grounding for the current analytical step.

Simultaneously, the invariant Action Directive Guidance ($\mathbf{g}_v$) serves as a strict procedural control signal. Instead of being relegated to the background context, it is injected directly into the active reasoning stream. Mathematically, this is realized by appending the directive to the ongoing reasoning state: $\mathbf{r}'_{<t} = [\mathbf{r}_{<t} \oplus \mathbf{g}_v]$. This physical injection explicitly enforces the conditional dependency $P_\theta(\mathbf{s}_t \mid \mathbf{x}', \mathbf{r}'_{<t})$, ensuring that the model's immediate next generation step is rigidly constrained by the expert protocol rather than its own unguided extrapolation.

By disentangling the delivery of the "instruction of how to reason" ($\mathbf{g}_v$) from the "evidence for reasoning" ($\mathcal{C}_v$) through these distinct pathways, the IRG mechanism effectively mitigates context confusion. It ensures that the model remains structurally steered by the rigid SOP while being contextually informed by the dynamic knowledge base.

# 4. Experiment

## 4.1. Experiment Setup

### 4.1.1. DATASET

To comprehensively evaluate our framework, we selected a diverse suite of five reasoning tasks spanning two categories.

**1. Specialized, Expert-Driven Domains**: This category features three tasks characterized by their reliance on structured, multi-step expert workflows.

**Merchant Fraud Analysis** A financial task using a proprietary, real-world dataset to assess a merchant's fraud risk based on their information. This dataset will not be publicly released to comply with data privacy agreements.

**Medical Diagnosis and Astronomical Judgment** Two tasks sourced from the KUMO benchmark (Lin et al., 2025), which involve diagnosing diseases from patient results and identifying celestial objects from observations, respectively.

**2. General-Purpose Reasoning Benchmarks**: We also assess TG-RAG on two standard benchmarks to test fundamental reasoning abilities: GSM8K (Cobbe et al., 2021) for multi-step mathematical reasoning and StrategyQA (Geva et al., 2021) for multi-hop compositional reasoning.

### 4.1.2. IMPLEMENTATION DETAILS

We employ Qwen3-32B (Qwen Team, 2025) and the R1 Distill series (DeepSeek-AI, 2025) as our primary backbone models. All experiments are conducted on NVIDIA A100-80GB GPUs. To support the sequence interruption, the model inference is deployed utilizing the vLLM framework (Kwon et al., 2023). For the generation hyperparameters, we set the temperature to 0.7 and `top_p` to 0.95 to balance response diversity with strict procedural adherence. To ensure statistical reliability and address the inherent stochasticity of Large Reasoning Models, we execute 5 independent runs for each experimental setting. We report the mean performance alongside the standard deviation, utilizing end-to-end Accuracy as the primary evaluation metric.

### 4.1.3. BASELINES

We benchmark TG-RAG against two primary categories of baseline methods to strictly evaluate its advantages over both unconnected internal reasoning and existing retrieval-augmented paradigms.

**1. Non-Retrieval Baselines**: This category encompasses methods that rely solely on the LRM's parametric knowledge and in-context learning capabilities. Specific methods include:

- **Direct Reasoning**: Standard zero-shot prompting.

- **Few-Shot CoT (Few-CoT)** (Wei et al., 2022): Chain-of-Thought prompting with domain-specific demonstrations.

- **CoT-SC** (Wang et al., 2023c): Self-Consistency, which aggregates results from multiple reasoning paths to improve reliability.

- **Plan-and-Solve (PS)** (Wang et al., 2023b): A prompting strategy that explicitly guides the model to devise a plan before execution.

**2. Retrieval-Augmented Baselines**: We further bifurcate this category based on how the retrieved information is utilized:

- *Retrieval-Augmented Generation (RAG)*: These methods focus on retrieving factual domain knowledge to ground the generation. We compare against **Naive RAG** (Lewis et al., 2020), which employs a standard one-pass pre-retrieval strategy, and **Self-RAG** (Asai et al., 2024), a framework that utilizes reflection tokens to autonomously control retrieval demand and output quality.

- *Retrieval-Augmented Thought (RAT).*: These methods actively utilize retrieval to influence the reasoning trajectory itself. We include **IR-CoT** (Trivedi et al., 2023), which interleaves retrieval steps within CoT generation, and **RATT** (Zhang et al., 2025b), which uses retrieved information to revise thoughts.

## 4.2. Overall Comparison

The comparative results across 5 independent runs are summarized in Table 1. The data indicates that TG-RAG achieves consistent and competitive performance across a range of tasks and model architectures. A detailed examination reveals that the advantages of our framework are most pronounced in expert-driven domains, where adherence to complex procedural logic is critical.

In specialized domains such as Finance, Medical, and Astronomy, TG-RAG demonstrates a resilient performance advantage over the baselines. For instance, on the Finance task with the Qwen3-32B model, TG-RAG achieves a mean accuracy of 76.4%, surpassing Self-RAG's 64.1% and RATT's 71.2%. Similar trends are observed in the Medical domain with the R1-Distill-14B model, where TG-RAG (90.4%) outperforms both naive RAG (82.4%) and advanced IR-CoT (86.4%). Critically, the reported standard deviations ($\pm\sigma$) indicate that TG-RAG often maintains comparable or lower variance than baselines (e.g., $\pm 0.8$ vs. $\pm 1.3$ for IR-CoT on Qwen3/Astronomy), suggesting that the structured guidance of the EPG contributes to more stable and reproducible reasoning trajectories. This stability is particularly valuable in high-stakes professional applications where output consistency is as important as peak accuracy.

*Table 1.* Performance comparison of TG-RAG with baseline methods. TG-RAG demonstrates highly competitive or superior performance, with notable advantages in expert-driven domains.

| Model | Task | Non-Retrieval | | | | RAG | | RAT | | TG-RAG |
|---|---|---|---|---|---|---|---|---|---|---|
| | | Direct | Few-CoT | CoT-SC | PS | RAG | Self-RAG | IR-CoT | RATT | |
| R1-Distill-32B | Finance | $64.3_{\pm0.9}$ | $68.1_{\pm0.5}$ | $73.4_{\pm0.8}$ | $71.0_{\pm0.6}$ | $73.2_{\pm0.7}$ | $75.5_{\pm0.9}$ | $78.1_{\pm0.6}$ | $79.1_{\pm0.5}$ | $\mathbf{81.4_{\pm0.8}}$ |
| | Medical | $76.4_{\pm1.7}$ | $86.0_{\pm1.2}$ | $82.0_{\pm1.4}$ | $88.0_{\pm0.8}$ | $84.4_{\pm1.1}$ | $86.0_{\pm0.9}$ | $88.0_{\pm1.0}$ | $88.4_{\pm0.9}$ | $\mathbf{91.2_{\pm1.1}}$ |
| | Astronomy | $63.2_{\pm1.1}$ | $70.0_{\pm1.5}$ | $72.4_{\pm1.0}$ | $76.0_{\pm1.2}$ | $80.4_{\pm1.0}$ | $82.0_{\pm0.8}$ | $84.4_{\pm1.3}$ | $85.2_{\pm1.1}$ | $\mathbf{89.2_{\pm1.8}}$ |
| | GSM8K | $95.1_{\pm0.7}$ | $\mathbf{97.2_{\pm1.1}}$ | $96.5_{\pm0.6}$ | $95.3_{\pm0.8}$ | - | - | - | - | $96.1_{\pm0.7}$ |
| | StrategyQA | $91.2_{\pm0.7}$ | $92.1_{\pm0.5}$ | $93.4_{\pm0.4}$ | $91.5_{\pm0.6}$ | $90.2_{\pm0.8}$ | $92.0_{\pm0.7}$ | $91.8_{\pm0.9}$ | $93.1_{\pm0.4}$ | $\mathbf{94.2_{\pm0.7}}$ |
| R1-Distill-14B | Finance | $60.1_{\pm1.2}$ | $65.3_{\pm0.9}$ | $67.2_{\pm1.5}$ | $66.1_{\pm0.8}$ | $68.4_{\pm0.9}$ | $69.1_{\pm1.0}$ | $70.2_{\pm0.8}$ | $70.5_{\pm0.7}$ | $\mathbf{72.1_{\pm0.9}}$ |
| | Medical | $66.4_{\pm1.5}$ | $75.2_{\pm1.8}$ | $78.4_{\pm1.6}$ | $80.0_{\pm1.0}$ | $82.4_{\pm1.2}$ | $84.0_{\pm1.0}$ | $86.4_{\pm1.5}$ | $89.2_{\pm1.1}$ | $\mathbf{90.4_{\pm1.0}}$ |
| | Astronomy | $61.2_{\pm1.8}$ | $68.4_{\pm1.4}$ | $71.2_{\pm1.1}$ | $70.4_{\pm1.6}$ | $74.4_{\pm1.2}$ | $76.4_{\pm1.0}$ | $78.0_{\pm1.3}$ | $80.4_{\pm0.9}$ | $\mathbf{83.2_{\pm1.5}}$ |
| | GSM8K | $83.1_{\pm1.1}$ | $84.2_{\pm0.9}$ | $85.0_{\pm0.8}$ | $82.5_{\pm1.5}$ | - | - | - | - | $\mathbf{91.3_{\pm0.8}}$ |
| | StrategyQA | $84.2_{\pm0.9}$ | $85.1_{\pm0.7}$ | $86.4_{\pm0.6}$ | $85.3_{\pm0.8}$ | $85.5_{\pm1.0}$ | $86.2_{\pm0.8}$ | $87.1_{\pm0.9}$ | $87.5_{\pm0.5}$ | $\mathbf{89.1_{\pm0.7}}$ |
| Qwen3-32B | Finance | $51.2_{\pm1.4}$ | $59.4_{\pm1.8}$ | $54.1_{\pm2.1}$ | $60.5_{\pm1.2}$ | $62.3_{\pm0.9}$ | $64.1_{\pm1.1}$ | $67.5_{\pm1.3}$ | $71.2_{\pm0.8}$ | $\mathbf{76.4_{\pm1.0}}$ |
| | Medical | $84.0_{\pm0.8}$ | $88.4_{\pm0.6}$ | $88.0_{\pm0.7}$ | $78.4_{\pm1.5}$ | $86.4_{\pm0.9}$ | $88.0_{\pm0.8}$ | $88.4_{\pm0.7}$ | $90.0_{\pm0.6}$ | $\mathbf{91.6_{\pm0.8}}$ |
| | Astronomy | $90.2_{\pm0.6}$ | $92.0_{\pm0.7}$ | $\mathbf{94.4_{\pm0.4}}$ | $94.0_{\pm0.6}$ | $94.4_{\pm0.5}$ | $94.0_{\pm0.6}$ | $94.0_{\pm0.5}$ | $92.4_{\pm0.8}$ | $93.2_{\pm0.7}$ |
| | GSM8K | $95.3_{\pm0.6}$ | $97.0_{\pm0.4}$ | $96.2_{\pm0.5}$ | $96.1_{\pm0.6}$ | - | - | - | - | $\mathbf{99.1_{\pm0.2}}$ |
| | StrategyQA | $80.1_{\pm1.5}$ | $80.4_{\pm1.2}$ | $74.2_{\pm1.8}$ | $85.1_{\pm1.0}$ | $88.3_{\pm0.9}$ | $90.1_{\pm0.7}$ | $93.2_{\pm0.6}$ | $94.0_{\pm0.5}$ | $\mathbf{96.3_{\pm0.6}}$ |

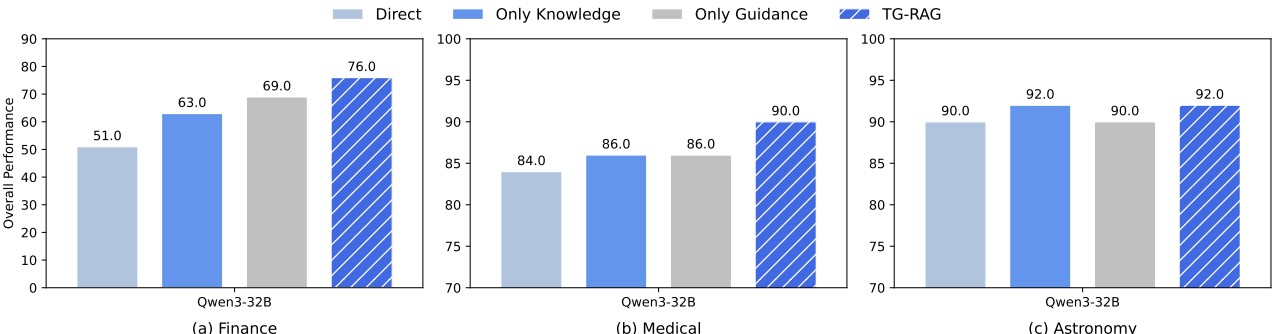

*Figure 4.* Performance Comparison of TG-RAG with and without Knowledge/Guidance Components.

The comparison with advanced Retrieval-Augmented Thought (RAT) methods further validates the efficacy of our "proactive steering" paradigm. While RAT methods (i.e., IR-CoT, RATT) reactively refine the model's self-generated thoughts, TG-RAG's injection of expert directives appears to be a more effective mechanism for enforcing procedural fidelity from the outset. This is evidenced by the consistent margins TG-RAG maintains over RATT in the Finance and Medical domain tasks across both R1-series and Qwen backbones.

For general-purpose reasoning benchmarks (GSM8K, StrategyQA), where the EPG is adapted into a linear Expert Procedure Chain (EPC), TG-RAG continues to show strong generalization capabilities. It is important to note that for GSM8K, retrieval-augmented baselines (RAG/RAT) were explicitly omitted from Table 1. As detailed in Appendix E, this exclusion ensures paradigm consistency. On StrategyQA, it consistently leads across all models (e.g., 96.3%

with Qwen3-32B). On GSM8K, TG-RAG achieves highly competitive results, reaching 99.1% with Qwen3-32B. We acknowledge that on GSM8K with the R1-Distill-32B model, Few-Shot CoT (97.2%) slightly surpasses TG-RAG (96.1%). This outcome is understandable, as GSM8K's relatively simpler procedural requirements may not fully leverage the specialized benefits of the EPG, allowing methods that rely on the model's intrinsic mathematical capabilities to perform comparably. Similarly, in the Astronomy task with Qwen3-32B, CoT-SC achieves a slightly higher mean accuracy (94.4% vs 93.2%). However, considering the overlapping confidence intervals implied by the standard deviations, these performances are statistically comparable.

Overall, the results indicate that TG-RAG effectively enhances reasoning in complex, logic-heavy specialized domains while remaining a robust and versatile tool for general reasoning tasks.

*Table 2.* The validation of the "Chain-of-Tree" Structure of EPG based on R1-Distill-32B.

| EPG Structure | Finance | Medical | Astronomy |
|---|---|---|---|
| Fixed-Path EPG | 0.76 | 0.86 | 0.80 |
| Chain-of-Tree EPG | **0.81** | **0.92** | **0.90** |

*Table 3.* Comparison of accuracy between Reasoning Process Injection and In-Prompt Integration strategies (Based on R1-Distill-32B).

| Integration Strategy | Finance | Medical | Astronomy |
|---|---|---|---|
| In-Prompt Integration | 0.72 | 0.90 | 0.78 |
| Reasoning Process Injection | **0.81** | **0.92** | **0.90** |

## 4.3. Ablation Studies

### 4.3.1. THE CHAIN-OF-TREE STRUCTURE OF EPG

To evaluate the necessity of the conditional branching inherent in our "Chain-of-Trees" architecture, we conducted an ablation study comparing the full EPG against a simplified Fixed-Path EPG variant. In this linear baseline, the topological structure was flattened: mutually exclusive branches were collapsed into a single, comprehensive guidance sequence containing instructions for all potential scenarios. This effectively creates a static, non-adaptive workflow, isolating the contribution of the dynamic routing mechanism.

The results, presented in Table 2, indicate a consistent performance advantage for the Chain-of-Trees architecture. TG-RAG outperforms the Fixed-Path variant across all three specialized domains, achieving notable accuracy gains of $5\%$ in Finance, $6\%$ in Medical, and $10\%$ in Astronomy. This performance gap suggests that a static reasoning structure is suboptimal when handling the conditional complexity characteristic of expert tasks.

The comparative underperformance of the Fixed-Path EPG can be attributed to information redundancy and increased cognitive load. By compelling the LRM to ingest a generic, inclusive procedure, the linear approach prevents the model from pruning irrelevant logical paths based on intermediate states. For instance, in medical diagnosis, the subsequent analysis is strictly contingent on specific test results (e.g., positive vs. negative). A linear workflow presenting instructions for both outcomes forces the model to filter through conflicting directives, introducing noise that may disrupt the reasoning focus. In contrast, the Chain-of-Trees structure excels by enabling dynamic path pruning. At each decision juncture, the IRG mechanism allows the model to lock onto context-dependent choices, effectively mirroring an expert's cognitive process of discarding irrelevant possibilities. This targeted guidance reduces the input noise, ensuring the reasoning trajectory remains precise and aligned with the evolving task state.

### 4.3.2. THE EFFECTIVENESS OF DIFFERENT COMPONENTS

To strictly evaluate the individual contributions of the two channels in our Dual-Pathway Injection mechanism (Section 3.3.2), we conducted a component ablation study. We established two decoupled variants: **Knowledge-Only**, which injects retrieved domain context via the input channel ($\mathcal{C}_v$) without procedural directives; and **Guidance-Only**, which appends action directives ($\mathbf{g}_v$) to the reasoning process without input augmentation.

The results, illustrated in Figure 4, indicate that the full TG-RAG framework, which integrates both pathways, consistently achieves the optimal performance across tasks. This suggests a synergistic effect where procedural structure and factual context mutually reinforce each other. The data further reveals distinct domain-dependent characteristics. In process-intensive domains like Finance, the performance gain from the full model is marked, exceeding the individual contributions of either isolated component. This highlights that complex analysis requires both "knowing the facts" and "knowing the steps." Conversely, in knowledge-intensive domains such as Astronomy, the Knowledge-Only variant already yields substantial improvements, reflecting the task's heavy reliance on factual recall. Notably, even in these fact-heavy scenarios, the full TG-RAG model matches or slightly exceeds the top performance, confirming that the imposition of procedural structure does not impede reasoning when factual retrieval is the primary driver.

In conclusion, these empirical findings validate the necessity of the dual-injection design. They suggest that procedural guidance and domain knowledge play distinct but complementary roles: guidance constrains the *logical trajectory*, while knowledge supplies the *informational substance*. The integration of both is therefore essential for achieving robust performance across diverse expert domains.

### 4.3.3. IMPACT OF THE INTEGRATION STRATEGY

A core distinction of our Dual-Pathway Injection mechanism is the separation of procedural signals from the passive input context. To validate this design, we conducted an ablation study comparing our "*Reasoning Process Injection*" strategy against a standard "*In-Prompt Integration*" baseline.

In the "*In-Prompt Integration*" baseline, the expert guidance ($\mathbf{g}_v$) retrieved from the EPG is treated as supplementary context. It is prepended to the user input $\mathbf{x}$ (i.e., $\mathbf{x}' = [\mathbf{g}_v, \mathbf{x}]$), relying on the model's instruction-following capability to adhere to the plan.

The results, presented in Table 3, indicate that "*Reasoning Process Injection*" consistently outperforms the "*In-Prompt*"

baseline across all domains. The performance gap is particularly striking in complex tasks like Astronomy, where our injection strategy achieves an accuracy of 0.90 compared to 0.78 for "*In-Prompt*" Integration—a substantial 12% improvement.

This disparity reveals the critical difference between passive context and active constraints. When guidance is placed in the prompt (In-Prompt), it competes with other informational tokens for attention and often suffers from attention dilution in long-context scenarios. In contrast, by injecting the guidance into the reasoning process, TG-RAG transforms the directive into a deterministic state transition. This physically forces the model's next-token prediction to strictly align with the expert protocol, ensuring the guidance is not merely "considered" as an external suggestion, but is "executed" as an active step of the reasoning chain.

### 4.3.4. THE IRG DYNAMIC INJECTION MECHANISM

This study disentangles the contribution of dynamic execution from the informational content provided by the EPG. We established a controlled comparison using a Template Prompting baseline, where the exact same procedural logic and domain knowledge are linearized into a static blueprint and provided entirely within the initial system prompt. Notably, this baseline essentially serves as a direct, domain-specific implementation of the Buffer of Thoughts (BoT) paradigm (Yang et al., 2024), which retrieves a comprehensive thought-template upfront and relies on the model's global instruction-following capabilities to guide the entire reasoning trajectory holistically.

The results in Table 4 reveal a marked contrast between these paradigms. TG-RAG consistently outperforms the static BoT-style baseline, achieving a substantial 14% accuracy gain in the Finance domain. This performance gap suggests that the static, upfront approach is undermined by contextual saturation—where the model struggles to parse instructions within a lengthy prompt—and adherence decay, where strict compliance with the plan diminishes as the generation length increases.

In contrast to the prompt-level, static intervention of BoT, the IRG mechanism operates at the inference-engine level, functioning as a dynamic attentional focusing mechanism. By delivering only the granular, immediate, and necessary guidance at each step, it simplifies the model's cognitive task from processing a complex global plan to executing a single local directive. Crucially, this step-by-step injection serves as a periodic corrective signal. By explicitly inserting the correct next procedural constraint, it effectively "re-anchors" the reasoning trajectory at every decision juncture. This mechanism prevents the accumulation of early-stage errors, empirically validating that active, step-wise structural injection provides far more robust procedural compliance in

*Table 4.* Ablation on Dynamic Execution: Comparing TG-RAG (Dynamic) with Template Prompting (Static) based on R1-Distill-32B.

| Method | Finance | Medical | Astronomy |
|---|---|---|---|
| Template prompting | 0.67 | 0.86 | 0.80 |
| TG-RAG (ours) | **0.81** | **0.92** | **0.90** |

complex workflows than static template retrieval.

## 5. Conclusion

In this study, we address Cognitive Drift, a phenomenon where Large Reasoning Models deviate from established Standard Operating Procedures (SOPs) during complex specialized domain tasks. We identify that standard retrieval-augmented approaches are often insufficient for such scenarios because they primarily focus on augmenting input context rather than enforcing logical consistency. To resolve this, we present TG-RAG, a framework that imposes explicit structural constraints on the model's inference process. By constructing an Expert Procedure Graph (EPG) to define valid logical topologies and employing the 'Interrupt-Retrieve-Generate' (IRG) mechanism, the framework injects expert directives directly into the ongoing reasoning process. This strategy transforms guidance from passive reference material into active state constraints, effectively preventing the model from wandering into invalid reasoning branches. Empirical evaluations across diverse specialized and general domains indicate that TG-RAG achieves consistent performance advantages over strong baselines.

## Acknowledgements

This work is supported by Ant Group. It is also partially supported by the Noncommunicable Chronic Diseases-National Science and Technology Major Project (NO. 2024ZD0532400 and NO. 2024ZD0532403) and the National Natural Science Foundation of China (No. 62372013)

## Impact Statement

This paper presents work whose goal is to advance the field of machine learning, specifically in enhancing the controllability of Large Reasoning Models for procedural tasks. While our framework aims to improve adherence to expert protocols in specialized domains, it serves as a research exploration rather than a deployment-ready solution. Consequently, human oversight remains essential, particularly in high-stakes decision-making scenarios where errors could have significant consequences.

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

## A. Empirical Analysis of Open Branch Activation

In Section 3.3.1, we introduced the Open Branch mechanism as a fallback strategy to handle Out-of-Distribution (OOD) scenarios. Here, we address the empirical reliability of this mechanism in practical deployments.

Our experiments across specialized domains (Finance, Medical, Astronomy) indicate that the Open Branch is triggered in less than 1% of reasoning steps. This low activation rate serves as a strong validation of the EPG construction quality. It aligns with the fundamental nature of Standard Operating Procedures: by definition, SOPs are designed to standardize high-frequency, normative workflows, covering the vast majority of operational scenarios. The open branch functions ensures system resilience against unforeseen anomalies without implying that the primary EPG structure lacks coverage for standard tasks.

## B. EPG Construction and Real-World Example

This section provides a detailed breakdown of the Expert Procedure Graph (EPG) construction procedure and presents a concrete visual example of its execution in a specialized task.

### B.1. Construction Procedure: Deterministic Topological Mapping

A critical consideration in formalizing domain expertise is mitigating the risk of structural hallucinations during the construction phase. Unlike generic text extraction tasks, the construction of an EPG leverages the inherent structure of specialized Standard Operating Procedures (SOPs). In domains such as medical diagnosis or financial compliance, SOPs are not amorphous text; they already feature explicit conditional logic, decision trees, and rigid evaluation checklists. Consequently, the EPG construction is formulated as a deterministic topological mapping rather than an open-ended generation task.

The semi-automated construction pipeline consists of three standardized steps:

1. **Parsing Structured Source Material:** We utilize a Large Reasoning Model as a structural parser to process the raw SOP documents. The LRM is prompted to identify key analytical stages, condition checks, and terminal actions defined within the protocol.

2. **Topological Mapping:** The identified procedural elements are deterministically mapped into graph components.

   - *Nodes (States):* Procedural states are instantiated as EPG nodes. The parser extracts the specific action required at that state to form the Action Directive Guidance ($\mathbf{g}_v$), and identifies the necessary background concepts or data requirements to define the Knowledge Container ($\mathcal{C}_v$).
   - *Edges (Routing):* Conditional requirements and decision thresholds (e.g., "If blood pressure $> 140/90$") are translated into mutually exclusive routing edges connecting the nodes.

3. **Expert Verification (Human-in-the-Loop):** The candidate graph undergoes a structural review by domain experts. Because the LRM merely transcribes existing explicit logic into a graph format, the expert's role is confined to verifying the topological correctness of decision branches (e.g., ensuring no missing edge conditions) rather than authoring content. This focused verification reduces human involvement to approximately 30 minutes per task class, creating a persistent logic base that amortizes this one-time cost across massive-scale inferences.

### B.2. Real-World EPG Structural Example

To concretely illustrate how the theoretical Expert Procedure Graph (EPG) is operationalized in code, Figure 5 presents a formalized textual representation of a Medical EPG.

In our implementation, the EPG is structured as a nested dictionary format. This allows for the precise definition of the "Chain-of-Trees" topology discussed in Section 3.2.

This code snippet directly maps to our mathematical formulation of the EPG:

- **Topology** (`child_type`): The framework utilizes `"sequential"` to define the macro-level analytical stages (the "Chain"), and `"choices"` to define the micro-level conditional routing (the "Trees"). The `choice_words` act as the routing edge conditions.

- **Node Attributes** (`cot_guide`): Each node contains a `cot_guide` dictionary representing the tuple $(\mathbf{g}_v, \mathcal{C}_v)$. The `words` key corresponds to the Action Directive Guidance ($\mathbf{g}_v$) injected into the sequence, while the `knowledge` key serves as the Knowledge Container ($\mathcal{C}_v$).

- **Reflection Nodes:** As seen in the `conclusion_hierarchy`, terminal nodes are explicitly programmed to force a summarization and a final self-correction reflection pass, mitigating upstream routing errors before the final output is finalized.

## C. Efficiency Analysis

A critical consideration for any iterative reasoning framework is inference latency. While TG-RAG introduces a measurable inference overhead, this is an acceptable and entirely justified trade-off in strict specialized domains (e.g., Finance, Medical) where procedural compliance and zero-hallucination are paramount. This section analyzes the latency footprint of TG-RAG, compares its algorithmic complexity with other control methods, and validates an effective lightweight routing optimization.

### C.1. Linear vs. Exponential Complexity

As detailed in Table 5, TG-RAG introduces latency overhead compared to single-pass methods (e.g., CoT, Direct Reasoning). This is an expected trade-off for the transition from free-form generation to state-guided navigation. However, compared to search-based methods like Tree of Thoughts (ToT), which suffer from combinatorial explosion and exponential cost, TG-RAG's overhead is strictly linear with respect to the depth of the reasoning chain. Furthermore, unlike "Generate-then-Revise" frameworks (e.g., RAT) that incur the cost of generating full erroneous chains before correcting them, TG-RAG employs a preventative routing mechanism. By investing computation in "thinking before routing," we avoid the larger latency penalty of generating long, invalid trajectories.

### C.2. Latency Breakdown and KV Cache Optimization

A granular breakdown of time consumption (Table 6) reveals that the latency is not uniformly distributed. The actual generation phases are remarkably efficient, benefiting from modern inference engine optimizations. Specifically, by leveraging the KV Cache, the re-processing of the shared reasoning prefix during the "Continue Generation" steps is nearly instantaneous.

Consequently, the overhead is almost entirely concentrated in the Branch Selection steps. When utilizing the massive primary LRM (Origin, 32B) as the semantic router, these selection steps account for the vast majority ($\sim 85\%$) of the total inference time. Crucially, this is not because the semantic routing task is inherently complex. Rather, it is an artifact of employing LRMs. LRMs natively operate on a comprehensive "think-reflect-answer" paradigm, causing them to "overthink" and generate unnecessarily long internal reasoning chains for simple, deterministic condition checks (e.g., verifying if a numerical value falls within a normal diagnostic range).

### C.3. Optimization Strategy: Lightweight Routing

The identification of this LRM artifact highlights a clear avenue for optimization. The routing task—essentially a multiple-choice classification problem checking deterministic conditions—does not require strong reasoning ability.

To demonstrate that this overhead is not an inherent bottleneck of the TG-RAG framework, we implemented a decoupled routing strategy by offloading the semantic routing task to a lightweight, non-reasoning model (Qwen2.5-7B). As shown in Table 5, this strategy reduces the overall time consumption by $> 50\%$ across all domains. Table 6 further demonstrates that individual branch selections now require only 4–5 seconds. Importantly, this massive efficiency gain is achieved without sacrificing routing accuracy, proving that smaller models are fully capable of handling these deterministic checks while preserving the high-quality reasoning of the main LRM for the actual step-generation phases.

## D. Detailed Prompt Description

This section provides detailed examples of the prompts used in our TG-RAG method, all based on the medical diagnosis task. Figure 6 shows the prompts for continuing the reasoning process after a guidance sequence is injected. The prompts for having the model select the most appropriate branch from a list of choices are detailed in Figure 7. Finally, when

*Table 5.* Comparison of time[s] consumption of different methods.

| Task | Direct | Few-CoT | CoT-SC | PS | TG(Origin, 32B) | TG(Offload, 7B) |
|---|---|---|---|---|---|---|
| Finance | 35 | 35 | 37 | 50 | 110 | 54($\downarrow 51\%$) |
| Medical | 28 | 28 | 29 | 45 | 90 | 42($\downarrow 53\%$) |
| Astronomy | 26 | 25 | 25 | 40 | 87 | 40($\downarrow 54\%$) |

*Table 6.* Specific time consumption case of medical diagnosis scenario.

| Step | Action | Time(Origin, 32B)[s] | Time(Offload, 7B)[s] |
|---|---|---|---|
| 1 | Initial Generation | 1 | 1 |
| 2 | Branch Selection 1 | **11** | **4** |
| 3 | Continue Generation 1 | 2 | 2 |
| 4 | Branch Selection 2 | **13** | **5** |
| ... | ... | ... | ... |
| | total time | 88 | ˜**43** |

no predefined option is suitable, the prompts shown in Figure 8 are used to have the model generate a new, open-ended guidance.

## E. Baseline Implementation and Knowledge Source Details

This section elaborates on the construction of knowledge sources for our baselines and clarifies the distinct knowledge integration mechanism within the TG-RAG framework.

### E.1. Knowledge Corpora for Baselines

For the retrieval-augmented baselines (RAG and RAT), we constructed task-specific corpora tailored to the nature of each benchmark:

- **General and Domain Knowledge (Medical, Astronomy, StrategyQA)**: For tasks requiring a synthesis of broad and specific facts, we compiled a comprehensive corpus by merging relevant Wikipedia articles (Wikimedia Foundation, 2025) with the ground-truth context provided within the datasets.

- **Specialized Knowledge (Finance)**: For the highly specialized Finance task, we created a bespoke knowledge base by distilling key insights and analytical principles from internal expert documentation.

### E.2. Exclusion of GSM8K from Retrieval Baselines

Notably, retrieval-based methods were omitted for the GSM8K benchmark. This decision was made to ensure paradigm consistency. The performance of RATT baseline on GSM8K relies on the Program-Aided Language (PAL) paradigm (i.e., retrieving and executing Python code). In contrast, TG-RAG and our selected baselines operate strictly within the Natural Language Chain-of-Thought (CoT) paradigm. We excluded code-execution strategies to rigorously evaluate the effectiveness of reasoning steering without the confounding variables introduced by external symbolic interpreters.

### E.3. Knowledge Integration in TG-RAG

In contrast to the large, unstructured text corpora used for baselines, TG-RAG performs retrieval over a highly structured, pre-compiled knowledge source—the Expert Procedure Graph (EPG). As detailed in Section 3.2, specific procedural guidance ($\mathbf{g}_v$) and supplementary knowledge containers ($\mathcal{C}_v$) are curated and directly coupled within each node during construction. Consequently, the retrieval process in TG-RAG is fundamentally a state-dependent retrieval from a structured topology. When the IRG mechanism routes to a specific node $v$, the pre-associated tuple $(\mathbf{g}_v, \mathcal{C}_v)$ is fetched deterministically. This implementation choice ensures that the retrieved information is not merely semantically similar to the query (as in vector-based RAG), but is the precise, expert-vetted knowledge strictly required for the current procedural step.

**Example of Expert Procedure Graph**

```
child_tree_1 ={
    "cot_guide":{
        "knowledge": "...",
        "words": "Now, let me check the patient's Blood Glucose Test result, the result shows",
        "max_words": 75
    },
    "child_type": "choices"
    "child":[
        {
            "choice_words": "test result can rule out Pre-Diabetes",
            cot_guide":{
                "words": "Accoding to the Blood Glucose Test result and medical knowledge, i can rule out Pre-Diabetes. So it is...",
                "max_words": 0,
            }
        },
        {
            "choice_words" "test result can not rule out Pre-Diabetes",
            "cot_guide":{
                "words": "The Blood Glucose Test result can not rule out Pre-Diabetes. If the follow-up test results..",
                "max_words":0
            }
        }
    ]
}
main_hierarchy={
    "child_type": "sequential",
    "child":[
        child_tree_1,
        ...,
    ]
}

conclusion_hierarchy ={
    "child_type":"sequential",
    "child":[
        {
            "cot_guide":{
                "knowledge":"...",
                "words": "Now, let me summarize the above analysis and select a disease from the list of possible diseases...",
                "min_words": 2048
            },
            "cot_guide":{
                "knowledge": "...",
                "words": "At last, i have made a diagnosis, let's reflect on whether the evidence is reliable. ",
                "min_words": 1024
            }
        }
    ]
}

Medical_EPG={
    "child_type": "squential",
    "child":[
        main_main_hierarchy,
        conclusion_hierarchy
    ]
}
```

*Figure 5.* The example of the expert procedure graph in the medical diagnosis task.

**System Prompt**

You are a medical consultant. Below is a list of possible diseases that a patient may have and the results of some of the tests he has done. In addition, we will provide you with relevant knowledge about these possible diseases. Please diagnose the most likely disease among the patient's possible diseases based on the following information.
# Thinking format
When you are thinking, please separate each step of your thinking process and wrap them with
<analysis> and </analysis>, for example:
<think>
<analysis>
First thinking step
</analysis>
...
</think>
Please analyze the patient's examination results based on the medical knowledge provided, and determine whether the examination results can rule out the corresponding disease.
The length of each step should be controlled within the {min_words} word count.
Please strictly follow the medical diagnostic guidelines provided during your thinking process.

**User Prompt**

# User data
{data_prompt}
# Medical diagnosis knowledge
{knowledge}
Please strictly follow the medical diagnostic guidelines provided during your thinking process

*Figure 6.* The format of the system prompt and user prompt used when the model continues reasoning after the injection of a guidance sequence.

**System Prompt**

Suppose the following is an analysis text by an analysis expert. Based on the analysis results, please select the option that matches the analysis description from the given options, output the serial number directly, and do not output other content.

**User Prompt**

# User data
{data_prompt}
# analysis text
{prefix}
# Options
{actions}

*Figure 7.* The format of the system prompt and user prompt used when the model selects the next branch based on the current reasoning state.

**System Prompt**

You are an expert analyst and a master of logical reasoning. Your task is to determine the single most logical next step in a complex reasoning process, based on the problem description and the reasoning so far.
You must NOT perform the step yourself. Your entire output should be a short, concise phrase (3-7 words) describing this next analytical step. Do not add any preamble, explanation, or quotation marks. Just provide the guidance of the next step.

**User Prompt**

# Original Problem
{query}
# Reasoning So Far
{history}
---
Given the analysis so far, what is the single most logical next step to continue the investigation?

*Figure 8.* The format of the system prompt and user prompt used when the model generate the guidance of next open node based on the current reasoning state.

