# OpenReview forum: "TG-RAG: A Retrieval-Augmented Framework for Reasoning Guidance in Specialized Domains"
_ICML.cc/2026/Conference — ICML 2026 spotlight_

### Official Review · Reviewer_vuUs · 2026-02-25

**Soundness:** 3
**Presentation:** 3
**Significance:** 2
**Originality:** 2
**Overall Recommendation:** 5
**Confidence:** 3

**Summary:**

The authors introduce TG-RAG, a mechanism to inject domain-specific directives to the model during generation to steer it for domain-specific use-cases. They validate using different benchmarks across the finance, medical and astronomy domain.

**Compliance With Llm Reviewing Policy:**

Affirmed.

**Final Justification:**

The authors addressed my concerns and I raised the score

**Key Questions For Authors:**

Questions mentioned in the weaknesses

**Limitations:**

yes

**Strengths And Weaknesses:**

## Strengths
- The method is simple, elegant and scalable. As mentioned by the authors, an LLM can be used to generate a preliminary EPG and have experts validate and correct details. This procedure can be scaled across different domains.
- The model's reasoning can be steered without additional fine-tuning making it cost-effective.
- The authors compare against the popular baselines and their evaluation seems complete.

# Weaknesses
- A critical component here seems to be the EPG construction procedure. I haven't seen anywhere in the paper the detailed procedure and I was wondering how trivial it is and how much subject to hallucinations is. I am not sure how easy is to go from a big unstructured document which contains tables/figures/unstructured text to an elegant graph.
- An example execution with real data would be important. I see appendix section B is empty.
- A very relevant paper that is not mentioned is Buffer of Thoughts [1]. In that work, reasoning templates are retrieved and the model is instructed to follow those templates. More comparison and discussion on this is needed

[1] Buffer of Thoughts: Thought-Augmented Reasoning with Large Language Models - NeurIPS 2024

---

> ### Author Rebuttal · Authors · 2026-03-30
>
> Many thanks to Reviewer vuUs for providing thorough and insightful comments.
> > **W1 & W2**: EPG Construction and Example Execution.
>
> We sincerely apologize for the empty Appendix B, which resulted from a LaTeX compilation error. We will fully restore the detailed case studies in the revision.
>
> Regarding the EPG construction, as outlined in Lines 190-197, we clarify that our target SOPs are fundamentally different from "big unstructured documents." The construction is not an open-ended knowledge extraction task, but rather a systematic, highly constrained procedure:
>
> 1. **Parsing Structured Source Material:** By definition, specialized SOPs (e.g., medical guidelines, financial compliance) already feature explicit conditional logic, decision trees, and rigid checklists.
>
> 2. **Deterministic Topological Mapping**: We directly translate these explicit rules into graph components—procedural states become nodes, and conditional requirements become routing edges. This is largely a deterministic mapping process rather than complex text generation.
>
> 3. **Hallucination Mitigation**: Because the mapping strictly follows the predefined rigid logic of the SOP and results in a highly compact graph, it inherently minimizes the risk of structural hallucinations and allows for highly efficient expert verification.
>
> The complete construction code for the Medical and Legal domains is available at: https://anonymous.4open.science/r/Thought-Guidance-081B/example/medical.py.
>
>
>
> > **W3:** BoT comparison and discussion
>
> We sincerely thank the reviewer for highlighting the highly relevant and pioneering Buffer of Thoughts (BoT) framework. We will add a detailed discussion of BoT in the revised related work. While both frameworks aim to enhance reasoning via structured templates/graphs, they adopt distinct design philosophies.
>
> **1. Technical Comparison**
>
> The fundamental difference lies in the granularity and integration mechanism of the guidance:
>
> - **Guidance Integration (Static One-off vs. Dynamic Step-wise)**: BoT retrieves a **comprehensive thought-template** and prepends it to the input prompt, leveraging the model's global instruction-following capabilities to guide the entire reasoning trajectory holistically. In contrast, TG-RAG dynamically retrieves and injects **granular, step-specific procedural rules** throughout the generation process.
>
> - **Intervention Level (Input-time Prompt vs. Inference-time Guidance)**: BoT operates primarily at the prompt level. TG-RAG operates at the inference-engine level via the IRG mechanism, physically interrupting the sequence generation to inject the next topological constraint.
>
> **2. Empirical Comparison**
>
> To empirically compare these two paradigms within our specialized scenarios, we refer to the "Template Prompting" baseline in Section 4.3.4. In this baseline, the entire EPG is converted into a comprehensive static blueprint and provided upfront in the prompt—which essentially serves as a direct, domain-specific implementation of the BoT paradigm.
>
> As demonstrated in our ablation study, while the BoT-style upfront template significantly improves over vanilla prompting, the TG-RAG dynamic injection achieves even higher procedural compliance across all tasks:
>
> Table: Empirical Comparison (Accuracy %)
> | Method Paradigm | Finance | Medical | Astronomy |
> | :--- | :--- | :--- | :--- |
> | Upfront Template (BoT Paradigm / Sec 4.3.4) | 0.67 | 0.86 | 0.80 |
> | TG-RAG | **0.81** | **0.92** | **0.90** |
>
> (Note: We will explicitly map this baseline to the BoT discussion in the revision).
> This empirical result validates that for highly constrained, multi-step SOPs, active step-wise injection provides stronger structural adherence than static template retrieval.

---

> > ### Author Rebuttal · Reviewer_vuUs · 2026-03-31
> >
> > Concerns fully resolved with clarifications and supporting experiments.

---

> > > ### Author Response · Authors · 2026-04-01
> > >
> > > We are sincerely grateful for your positive assessment and for raising your score! We truly appreciate your thoughtful recognition of our rebuttal, particularly regarding the clarifications on the EPG construction procedure and the comparison with Buffer of Thoughts. We will carefully incorporate your constructive feedback into our revision.
> > >
> > > Once again, thank you for your valuable insights. Your detailed comments have been immensely helpful to us!

---

### Official Review · Reviewer_1JwH · 2026-03-12

**Soundness:** 3
**Presentation:** 3
**Significance:** 3
**Originality:** 3
**Overall Recommendation:** 4
**Confidence:** 3

**Summary:**

The paper proposes TG-RAG (Thought Guidance-Retrieval Augmented Generation) to mitigate "Cognitive Drift"—where LRMs deviate from expert protocols during complex reasoning in specialized domains. It formalizes Standard Operating Procedures (SOPs) into an Expert Procedure Graph (EPG) using a "Chain-of-Trees" topology. The framework employs an Interrupt-Retrieve-Generate (IRG) mechanism to dynamically inject step-wise directives and factual knowledge directly into the reasoning process. Experiments in finance, medicine, and astronomy show that TG-RAG improves procedural adherence and accuracy over static prompting and traditional RAG.

**Compliance With Llm Reviewing Policy:**

Affirmed.

**Key Questions For Authors:**

1. How does the framework handle "SOP Incompleteness" beyond the fallback Open Branch? For instance, what if the model's self-evaluation for branch selection is itself wrong?


2. Table 5 shows a large latency overhead. Have you tested if a much smaller model (e.g., 1B-3B) can serve as the semantic router without sacrificing accuracy?


3. In Figure 4, why does "Only Knowledge" perform better than "Only Guidance" in Astronomy, while the opposite is true for Finance?

**Limitations:**

Yes

**Strengths And Weaknesses:**

- **Soundness**:

  - The "Chain-of-Trees" EPG structure logically accounts for conditional branching in expert tasks.

  - The dual-pathway injection (Guidance vs. Knowledge) effectively separates "how to reason" from "what information to use".

  - Empirical results show TG-RAG consistently leads in accuracy in specialized domains like Medical (90.4%–91.6%) and Finance (72.1%–81.4%).

- **Presentation**:

  - The concept of "Cognitive Drift" is well-defined and motivated.

  - Algorithm 1 and Figure 2 provide a clear roadmap of the IRG mechanism.

- **Significance**:

  - Addressing logical drift in safety-critical domains (medicine/finance) is highly relevant.

  - The system remains training-free and compatible with various LRMs.

- **Originality**:

  - The shift from "passive context" in standard RAG to "active steering" via injected reasoning states is a significant conceptual advancement.

**Weaknesses**:

-  **Latency**: TG-RAG significantly increases inference time (e.g., 110s vs 35s for Finance) due to repeated branch selection steps.





-  **Human Involvement**: While semi-automated, verifying the EPG topology still requires 30 minutes of expert time per domain, which may hinder large-scale scalability.





- **Routing Bottleneck**: Branch selection accounts for ~85% of total inference time, indicating the framework heavily depends on the efficiency of the semantic router.

---

> ### Author Rebuttal · Authors · 2026-03-30
>
> Many thanks to Reviewer 1JwH for insightful comments.
> > **W1, W3, Q2:** Time Consumption and Efficiency
>
> We emphasize that TG-RAG achieves an outstanding trade-off between procedural effectiveness and computational efficiency. Specifically:
>
> 1. **An acceptable trade-off for high-stakes accuracy.** Inference overhead is justified in specialized domains (e.g., Finance, Medical) where procedural compliance and zero-hallucination are paramount. Extra time translates directly to robust error mitigation and SOP adherence.
>
> 2. **The overhead is an LRM artifact, not an inherent routing bottleneck.** Branch selection accounts for most inference time, not due to routing complexity, but because LRMs' "think-reflect-answer" paradigm causes "overthinking"—generating unnecessarily long reasoning chains for simple deterministic condition checks. (e.g., verifying if a value falls within a normal range).
>
> 3. **A lightweight non-reasoning router substantially reduces this overhead.**
> Offloading routing to a lightweight, non-reasoning model (Qwen2.5-7B) reduces overall time by >50% across domains (Table 1), with individual selections taking only 4–5s (Table 2). This massive efficiency gain sacrifices zero routing accuracy, proving smaller models easily handle deterministic checks.
>
> Table 1:Time Consumption Comparison (s)
> | Task | Direct | TG-RAG (Origin, 32B) | TG-RAG (Offload, 7B) | Decrease |
> | :--- | :--- | :--- | :--- | :--- |
> | Finance | 35 | 110 | 54 | 51% |
> | Medical | 28 | 90 | 42 | 53% |
> | Astronomy | 26 | 87 | 40 | 54% |
>
> Table 2: Time Breakdown with 7B Router (Medical Scenario)
> | Step | Action | Time (s) |
> | :--- | :--- | :--- |
> | 1 | Initial Generation | 1 |
> | 2 | Branch Selection 1 | 4 |
> | 3 | Continue Generation 1 | 2 |
> | 4 | Branch Selection 2 | 5 |
> | | Total Time | ~43 |
>
> We will integrate this lightweight routing strategy and the efficiency analysis into the revision.
>
> > **W2:** Human involvement
>
> We would like to clarify that this initialization step is highly compatible with large-scale deployment due to three practical considerations:
>
> 1. **One-Time Upfront Cost**: Expert verification is a single, domain-specific initial setup, not a query-specific recurring query cost.
>
> 2. **Massive-Scale Amortization**: SOPs address static, standardized workflows. Once an EPG is validated, it serves as a persistent logic base capable of executing massive-scale inferences with zero additional human intervention. Thus, the 30-minute effort is effectively amortized across thousands of subsequent queries.
>
> 3. **Efficient Maintenance**: When domain protocols undergo periodic updates, the framework permits quick, localized node modifications on the existing EPG, requiring negligible time compared to building from scratch.
>
> > **Q1:** SOP incompleteness
>
> Regarding SOP incompleteness: Domain SOPs are inherently comprehensive. The human-in-the-loop expert verification during construction prevents structural omissions. For rare edge cases, a fallback "Open Branch" gracefully degrades into standard LRM reasoning, preventing failure.
>
> Regarding incorrect branch selection (routing errors), the framework mitigates catastrophic failure through two architectural designs:
>
> - **Error Isolation via Forest Structure**: The EPG is structured as a collection of decoupled decision trees. An incorrect routing decision in one sub-tree is confined to that specific local analysis, preventing the error from cascading into independent, subsequent logic branches.
>
> - **Mandatory Reflection Nodes**: We explicitly embed terminal reflection nodes at the end of key EPG trajectories. This forces the model to conduct a final self-correction pass, reviewing the generated reasoning trace against the initial query to detect and rectify preceding routing errors.
>
> > **Q3:** Domain Characteristics
>
> The varying performance between "Only Knowledge" and "Only Guidance" reflects the distinct reasoning requirements of different domains:
>
> - **Astronomy (Fact-Intensive)**: These tasks rely heavily on specific physical constants and mathematical formulas. Without retrieving this exact factual data ("Knowledge"), the model cannot derive the correct answer regardless of the reasoning structure. Thus, explicit knowledge retrieval acts as the primary performance bottleneck.
>
> - **Finance (Procedure-Driven)**: Financial reasoning (e.g., fraud analysis) is strictly rule-based. The necessary data (e.g., transaction records) is typically already present in the input context. The main challenge is preventing the model from skipping mandatory compliance verification steps. Therefore, structural constraints ("Guidance") are far more critical than retrieving external facts.
>
> This domain-specific divergence highlights the necessity of TG-RAG's dual-pathway design, which accommodates both factual and procedural requirements. We will add this analysis to the Figure 4 discussion in the revised manuscript.

---

### Official Review · Reviewer_1Sqq · 2026-03-12

**Soundness:** 4
**Presentation:** 3
**Significance:** 3
**Originality:** 2
**Overall Recommendation:** 5
**Confidence:** 3

**Summary:**

The paper proposes the Thought Guidance-Retrieval Augmented Generation (TG-RAG) framework for implementing complex workflows involving language models in specialized domains to avoid cognitive drift. The framework relies on Expert Procedure Graphs (EPG) that formalize unstructured Standard Operating Procedures (SOP), where each node contains instructions and knowledge context, which can either be static facts or can be retrieved via RAG. An EPG, inspired by decision trees, is a usually a static nested chain-based reasoning topology, where each node is a tree. EPGs can be AI generated with additional human feedback and knowledge context can be updated with the latest data without any structural changes to the topology itself. The EPG is then used by a dynamic Interrupt-Retrieve-Generate (IRG) mechanism during reasoning generation to inject directions specific to the current reasoning step into the generation process. Specifically the mechanism identifies the current state within the high-level chain, then determines the next direction based on that state in the respective decision tree, before continuing the generation. This selection is done with an RLM with semantic routing and an open branch, i.e. no specific directions, as a fallback mechanism, which is however rarely triggered (less then 1%). This procedure aims to influence the generation more actively, then just providing a plan as passive contextual information, so that rule-following is more enforced, i.e. that instructions are not ignored and also to reduce hallucination in multi-step reasoning.

**Compliance With Llm Reviewing Policy:**

Affirmed.

**Final Justification:**

The rebuttal resolved my concerns and will hopefully improve the camera-ready version. I will keep my score of accept.

**Key Questions For Authors:**

1. How is the EPG selected? (From Algorithm 1 I assume by the user alongside the query.)
2. How/when is the IRG triggered? (From the appendix, I assume that the RLM is actually prompted to generate individual thoughts, i.e., reasoning steps.)

**Limitations:**

yes

**Strengths And Weaknesses:**

strengths:
* addresses a relevant topic: aligning reasoning/workflow handling in specialized domains
* for the most part clearly written
* technical sound
* nice experiments and justification of baselines
  * good ablation study

weaknesses:
* certain important details such as the EPG selection and how or when the IRG is triggered are not discussed (see also questions)
* things like missing author names in the references (see below) are also a bit concerning in regards to rigor in scientific writing
* additionally I have some concerns regarding the novelty: one could model the EPGs as Graph of Thoughts. While the original framework by Besta et al. only indirectly supports conditional statements to implement decision trees and does not support RAG, additional work, that generalizes reasoning operations, including RAG, was later proposed in [1].

other issues:
* Appendix B is empty besides the title
* references:
  * while consistently applied, there is not place of publication for arXiv references, which can only be gleaned from the URL
  * [Foundation W.]:
    * such "author" names should probably be encapsulted with additional brackets (`{}`), so that are readable
    * missing year, missing accessed date
  * [Yang and Li 2025] - other authors are missing or at least an "et al."
    * also [Wei and Wang 2022] and potentially others
  * [Zhang et al. 2025a/2026] - URLs cut into the second column, for example use the LaTeX package `xurl` to avoid this
  * titles: you might want to capitalize at least proper names and abbreviations such as "AI"
* minor issues:
  * 2.1, line 084, second column: "steering.+" - remove plus sign
  * 3.2, line 181, first column: "figure 3" - figure should be capitalized in this context
  * 4.2, second ("In specialized domains...") and third ("This stability is...") paragraphs should probably be merged into a single paragraph.
  * 5, line 430, second column: "cess.By" - missing whitespace
  * Appendix D:
    * after each paragraph title, there is no whitespace before the subsequent sentence
    * line 649: "errors.If" - missing whitespace
    * line 658: "protocols.By" - missing whitespace
    * line 668: "exponentially.TG-RAG" - missing whitespace
  * Tables 5 and 6 are missing a unit

[1] Besta et al. (2025) Reasoning Language Models: A Blueprint arXiv:2501.11223.

I was torn between a weak accept and an accept, but in the end choose the latter to give the authors the benefit of the doubt in addressing the issues above.

---

> ### Author Rebuttal · Authors · 2026-03-30
>
> We sincerely thank Reviewer 1Sqq for the detailed review and insightful questions!
> > **W1&Q1:** EPG Selection
>
> As detailed in Section 3.2, the EPG is predefined for each benchmark/domain, not dynamically selected per query. This choice is based on two primary considerations:
>
> 1. **Alignment with Domain SOPs**: EPGs formalize domain-specific SOPs (e.g., financial fraud analysis). For a given class of specialized tasks, the underlying compliance rules and workflows are universal and highly stable. A fixed EPG accurately reflects the persistent nature of these expert protocols.
>
> 2. **Error Prevention**: In high-stakes domains, topological correctness is paramount. Dynamically selecting an EPG per query introduces unnecessary risks of routing errors or retrieval hallucinations.
>
> We will explicitly clarify this static evaluation setup in the revised methodology section.
>
>
> > **W1&Q2:** IRG Trigger
>
> We would like to clarify that the IRG triggering mechanism (Section 3.3.2) is achieved through a three-step pipeline:
>
> 1. The mechanism is triggered precisely at **the completion of each discrete reasoning step**, the LRM is prompted to encapsulate each individual logical deduction within specific tags (i.e., <analysis> and </analysis>).
>
> 2. We leverage vLLM's sequence interruption mechanism to actively pause generation the exact moment the closing </analysis> tag is emitted.
>
> 3. During this suspended state, the semantic router evaluates the generated thought against the EPG, retrieves the procedural directive, injects it, and resumes generation.
>
> The exact implementation can be verified within our provided anonymous repository: https://anonymous.4open.science/r/Thought-Guidance-081B/thought_guidance.py.
>
> > **W3:** Novelty: TG-RAG vs. GoT
>
> We thank the reviewer for pointing out the comprehensive blueprint [1]. We agree that structurally, an EPG can be viewed as a graph modeling reasoning thoughts. However, TG-RAG differs fundamentally from the original GoT framework (Besta et al.) in both its topological origin and reasoning paradigm:
>
> - **Topology Origin (Ad-hoc Generation vs. Expert Formalization):** In GoT, the graph topology is generated **ad-hoc** by the LLM during inference, which introduces unpredictable structural variance. In contrast, our EPG **formalizes** verified domain SOPs, ensuring topological correctness and structural reliability.
>
> - **Reasoning Paradigm (Trial-and-Error Search vs. Procedural Steering):** GoT utilizes graphs for unbounded trial-and-error search to discover potential solutions. Conversely, TG-RAG utilizes the EPG for deterministic procedural steering, safely guiding the model's trajectory and preventing risky reasoning deviations in specialized domains.
>
> Furthermore, regarding the incorporation of RAG as proposed in [1], TG-RAG is not simply a straightforward combination of GoT and RAG. It introduces distinct design choices for both the retrieval target and the integration mechanism, as summarized below:
>
> Table: Comparison of RAG Integration Mechanisms
> | Dimension | Heuristic Search + RAG (e.g., [1]) | TG-RAG (Ours) |
> | :--- | :--- | :--- |
> | Retrieval Target | Factual: Background knowledge. | Procedural: Step-specific SOP directives. |
> | Integration Mechanism | Prompt Augmentation: Prepended to input. | IRG Mechanism: Injected into the reasoning sequence. |
>
> Specifically, we clarify these mechanistic distinctions:
>
> - **Retrieval Target (Factual vs. Procedural)**: When frameworks like [1] employ RAG, the typical objective is to retrieve **factual background knowledge**. In TG-RAG, the retrieval target is **step-specific procedural guidance** explicitly defined within the EPG.
>
> - **Integration Mechanism**: Generalized reasoning frameworks typically integrate retrieved content (or templates) by appending them to the **input prompt**. Unlike this approach, TG-RAG introduces the IRG mechanism (Section 3.3.2). IRG actively interrupts the inference processand injects the retrieved directives directly into the model's **ongoing reasoning sequence**. This physical intervention forces the model to condition its immediate next step directly on the injected constraint, achieving a rigid procedural steering.
>
> We will explicitly clarify these mechanistic distinctions and reference [1] in the revised related work.
>
> > **W2:** Formatting and Typos
>
> We sincerely thank the reviewer for the meticulous proofreading. We have corrected all identified formatting issues, including reference details (missing authors/venues, capitalization, URL overflows), text typos (stray characters, missing whitespaces), and paragraph merging.
>
> Regarding the empty Appendix B (caused by a compilation error), we apologize for the omission. We have provided a detailed real-data execution example in our response to Reviewer vuUs, and will fully restore this section in the revision.
>
> Finally, the time unit in Tables 5 and 6 is indeed seconds ("Time (s)" in the headers). We will highlight this more prominently in the captions.

---

> > ### Author Rebuttal · Reviewer_1Sqq · 2026-04-04
> >
> > Thank you for the rebuttal and the answers to my concerns.
> >
> > >In GoT, the graph topology is generated ad-hoc by the LLM during inference, which introduces unpredictable structural variance.
> >
> > That is not true. GoT also uses static graphs, specific for each task in their case. However while TG-RAG operates at the generation-level of a single model inference, GoT uses multiple distinct, potentially shorter, model inferences, so there are some obvious differences.
> >
> > >Regarding the empty Appendix B (caused by a compilation error), we apologize for the omission. We have provided a detailed real-data execution example in our response to Reviewer vuUs
> >
> > I could not find this example.
> >
> > After reading the other reviews and considering the rebuttal, I will maintain my score, as it is already high. I wish the authors good luck with their submission.

---

> > > ### Author Response · Authors · 2026-04-06
> > >
> > > Thank you for the follow-up and for maintaining your positive score. We would like to address the two points you raised.
> > >
> > > >  Comparison with GoT
> > >
> > > We acknowledge our misunderstanding regarding GoT's topology in the previous response. You are correct that GoT also utilizes static graphs. The core distinctions between GoT and TG-RAG are twofold:
> > >
> > > - **Inference Granularity**: As you pointed out, GoT relies on **multiple distinct model inferences**. In contrast, TG-RAG operates at **the sequence-generation level of a single model inference** via our IRG mechanism.
> > >
> > > - **Reasoning Paradigm**: GoT utilizes graph structures for trial-and-error search to discover potential solutions. Conversely, TG-RAG utilizes the EPG for deterministic procedural steering, strictly guiding the generation trajectory along established workflows.
> > >
> > > We will correct this comparison in the revised manuscript to reflect these precise distinctions.
> > >
> > > > Missing Appendix B Example
> > >
> > > Due to the text-only limitations of the rebuttal interface, we originally provided the EPG construction code to Reviewer vuUs instead of a visual execution graph, but missed updating the cross-reference in our response to you. For your reference, the code is available at: https://anonymous.4open.science/r/Thought-Guidance-081B/example/medical.py
> > >
> > > To address this, we provide a text-based snippet of the real-data execution example below:
> > >
> > > ```
> > > child_tree_1 ={
> > >     "cot_guide":{
> > >         "knowledge": "...",
> > >         "words": "Now, let me check the patient's Blood Glucose Test result, the result shows",
> > >         "max_words": 75
> > >     },
> > >     "child_type": "choices"
> > >     "child":[
> > >         {
> > >             "choice_words": "test result can rule out Pre-Diabetes",
> > >             cot_guide":{
> > >                 "words": "Accoding to the Blood Glucose Test result and medical knowledge, i can rule out Pre-Diabetes. So it is...",
> > >                 "max_words": 0,
> > >             }
> > >         },
> > >         {
> > >             "choice_words" "test result can not rule out Pre-Diabetes",
> > >             "cot_guide":{
> > >                 "words": "The Blood Glucose Test result can not rule out Pre-Diabetes. If the follow-up test results..",
> > >                 "max_words":0
> > >             }
> > >         }
> > >     ]
> > > }
> > > main_hierarchy={
> > >     "child_type": "sequential",
> > >     "child":[
> > >         child_tree_1,
> > >         ...,
> > >     ]
> > > }
> > >
> > > conclusion_hierarchy ={
> > >     "child_type":"sequential",
> > >     "child":[
> > >         {
> > >             "cot_guide":{
> > >                 "knowledge":"...",
> > >                 "words": "Now, let me summarize the above analysis and select a disease from the list of possible diseases...",
> > >                 "min_words": 2048
> > >             },
> > >             "cot_guide":{
> > >                     "knowledge": "...",
> > >                     "words": "At last, i have made a diagnosis, let's reflect on whether the evidence is reliable. ",
> > >                     "min_words": 1024
> > >                 }
> > >         }
> > >     ]
> > > }
> > >
> > > Medical_EPG={
> > >     "child_type": "squential",
> > >     "child":[
> > >         main_main_hierarchy,
> > >         conclusion_hierarchy
> > >     ]
> > > }
> > > ```
> > >
> > > The fully formatted case studies will be included in the camera-ready version. Thank you again for your time and constructive feedback.

---

### Decision · Program_Chairs · 2026-04-30

**Decision:**

Accept (spotlight)

**Comment:**

TG-RAG proposes an elegant training-free framework that actively steers LLM reasoning via an Expert Procedure Graph and an Interrupt-Retrieve-Generate mechanism, shifting from passive context augmentation to active procedural injection. Reviewers consistently acknowledge the method's simplicity, scalability, and strong empirical results across finance, medical, and astronomy domains. The ablation study effectively validates the dual-pathway design. Concerns about inference latency, EPG construction details, and novelty relative to prior graph-based reasoning frameworks were raised but adequately addressed during rebuttal, with the authors demonstrating a lightweight router that halves latency and providing empirical comparisons to template-based alternatives.